# Development of a Modified QuEChERS Method Based on Magnetic Multi-Walled Carbon Nanotubes as a Clean-Up Adsorbent for the Analysis of Heterocyclic Aromatic Amines in Braised Sauce Beef

**DOI:** 10.3390/foods12010138

**Published:** 2022-12-27

**Authors:** Min Li, Pengxiang Wang, Xu Zhang, Hongyu Wang, Ke Li, Yanhong Bai

**Affiliations:** 1College of Food and Bioengineering, Zhengzhou University of Light Industry, Zhengzhou 450001, China; 2Henan Key Laboratory of Cold Chain Food Quality and Safety Control, Zhengzhou 450001, China

**Keywords:** heterocyclic aromatic amines, QuEChERS, Fe_3_O_4_-MWCNTs, braised sauce beef

## Abstract

Heterocyclic aromatic amines (HAAs) generated during the cooking of meats cause adverse effects on human health. The purpose of the current research was to develop a modified QuEChERS (Quick, Easy, Cheap, Effective, Rugged, Safe) method using magnetic multi-walled carbon nanotubes (Fe_3_O_4_-MWCNTs) as clean-up adsorbents for the rapid determination of HAAs in braised sauce beef. The significant parameters in extraction and clean-up processes were screened and optimized. Under optimal conditions, the LODs ranged from 3.0 ng/g to 4.2 ng/g. The recoveries (78.5–103.2%) and relative standard deviations RSDs (<4.6%) of five HAAs were obtained. These are in accordance with the validation criteria (recovery in the range of 70–120% with RSD less than 20%). Compared with conventional clean-up adsorbents (PSA or C18), Fe_3_O_4_-MWCNTs displayed equivalent or better matrix removal efficiency, while making the pretreatment process easier and more time-saving through magnetic separation. Less usage of adsorbent makes the method possess another advantage of being lower in cost per sample. The method developed was successfully applied to analyze real samples collected from local deli counters, demonstrating Fe_3_O_4_-MWCNTs could be considered as an effective alternative adsorbent with great potential in the QuEChERS process.

## 1. Introduction

Heterocyclic aromatic amines (HAAs), one family of chemical substances with a heterocyclic structure, are mainly produced from high-protein foods (seafood and meat products) which are cooked at high temperatures. Currently, over 30 HAAs have been found and structurally identified in different cooked foodstuffs [1,2]; several of them are considered to be highly potential carcinogens and mutagens, to which humans are primarily exposed through diet [3,4]. Thus, the food safety issues of HAAs have received much attention by both consumers and researchers.

Currently, various analytical methods, including gas chromatography–mass spectrometry (GC–MS), liquid chromatography–mass spectrometry (LC–MS), liquid chromatography with fluorescence detection (LC–FLD) and/or diode array detection (LC–DAD), have been commonly employed to analyze HAAs [5,6,7]. Due to the trace amounts of HAAs and the complex food matrix, an appropriate sample pretreatment procedure before analysis is required. Currently, the most frequently used pretreatment methods for HAA analyses are solid-phase extraction (SPE) and tandem SPE [8,9], which are tedious, time-consuming, require a large number of reagents and chemicals, and generally offer a low recovery. Subsequently, many laboratories have also proposed other faster and more convenient strategies. Feng et al. (2022) established a method of magnetic solid-phase extraction (MSPE) based on novel magnetic covalent organic polymers for the detection of HAAs in fish and meat products [10]. Chevolleau et al. (2020) developed a method that combines SALLME extraction and SPE purification for the analysis of HAAs in cooked beef [11]. Recent studies (Hsiao et al., 2017; Chiang et al., 2022; Lai et al., 2023) have developed the QuEChERS method using a combination of PSA, MgSO_4_ and C18EC as the clean-up adsorbents to analyze HAAs from meat products, and proved that the QuEChERS method is a simple, rapid and convenient sample pretreatment method with broad applicability [12,13,14].

These outstanding superiorities of QuEChERS make it a popular technique in the analysis of contaminants in complex food samples, including HAAs. It includes two processes, extraction and clean-up [15]. The key step of the QuEChERS method lies in its clean-up process, and the clean-up adsorbent is an essential factor. The commonly used clean-up adsorbents include C18, graphitized carbon black (GCB) and primary secondary amine (PSA) [16]. PSA with primary and secondary amino groups could form strong hydrogen bonds with some matrix components. Thus, it exhibits the function of removing organic acids, some sugars and fatty acids. C18 has a strong adsorption capacity for non-polar interfering substances and GCB is mainly applied to remove pigments. However, when C18, PSA or GCB is used alone, the clean-up performance is not satisfactory for the treatment of complex matrix samples, and low sensitivity and recovery may occur. Therefore, these traditional adsorbents usually need to be used in combination, and the combination of multiple clean-up adsorbents made experimental procedures more tedious [17]. In order to solve the problems, the development of more efficient clean-up adsorbents is imperative. Moreover, high-speed centrifugation is inevitably used to separate the clean-up adsorbent from extracting solution after purification in the conventional QuEChERS approach, which makes the pretreatment process cumbersome and time-consuming.

Recently, multi-walled carbon nanotubes (MWCNTs) have attracted substantial interest in sample pretreatment, due to their unique hollow tubular structures, high surface area, and π-π stacking interactions; these features enable them to exhibit exceptional capabilities of matrix purification [18]. Nevertheless, when MWCNTs are served as clean-up adsorbents, the major challenge is their separation and recovery from water-based solutions. The magnetic functionalization of MWCNTs with Fe_3_O_4_ magnetic nanoparticles could achieve rapid separation and collection of Fe_3_O_4_-MWCNTs via a permanent magnet, which endows clean-up adsorbents with new capabilities, and preserves the inherent advantages of MWCNTs [19]. Compared with the classical QuEChERS process, magnetic separation needs no additional centrifugal devices, making the QuEChERS procedure more convenient and time-saving.

Braised sauce beef, namely “Jiang Niu Rou”, is traditionally served with ready-to-eat meat products. It is popular in some Asian countries, including China. Although braised sauce beef is processed at a relatively low temperature, and is often repeatedly boiled for a long time together with “Lao Lu”, namely the marinating juice [20]. Conventionally, the marinating juice is often used for several or dozens of cooking cycles, for even several months or years, which offers good flavor to braised sauce beef. However, the amount of precursors in the recycled marinating juice, such as glucose, creatine and free amino acids, increases significantly with increasing cooking cycles, facilitating the formation of various hazardous substances, such as HAAs [21]. Zhou et al. [8] discovered that several HAAs were found in braised sauce beef, and the content of HAAs increased with repeated cooking times of beef. Therefore, more attention should be paid to the monitoring and measuring of HAAs in braised sauce beef.

This study aimed to describe a modified QuEChERS technique based on Fe_3_O_4_-MWCNTs coupled with HPLC to simultaneously determine 2-amino-1-methylimidazo [4,5-b]quinoline (IQ [4,5-b]), 1-methyl-9*H*-pyrido [3,4-b]indole (Harman), 2-amino-3-methylimidazo [4,5-f]quinoline (IQ), 9*H*-pyrido [3,4-b]indole (Norharman) and 2-amino-5-phenylpyridine (Phe-P-1) in braised sauce beef. The experimental parameters in the modified QuEChERS process were optimized. The proposed technique was validated on the basis of its analytical performance, and finally applied to analyze real samples of braised sauce beef. This study not only offers a novel way for analyzing HAAs, but also has practical application values for routine monitoring in food safety.

## 2. Materials and Methods

### 2.1. Reagents and Materials

HAA standards, including 9*H*-pyrido [3,4-b]indole (Norharman), 2-amino-3-methylimidazo [4,5-f]quinoline (IQ), 1-methyl-9*H*-pyrido [3,4-b]indole (Harman), 2-amino-5-phenylpyridine (Phe-P-1) and 2-amino-1-methylimidazo [4,5-b]quinoline (IQ [4,5-b]), were ordered from Toronto Research Chemicals (North York, Ontario, Canada); their chemical structures are presented in Appendix A. All of the standards had a purity > 98%. All HPLC-grade reagents used in the HPLC system, including acetonitrile (ACN), methanol, ammonium acetate and acetic acid, were ordered from Sigma-Aldrich (Shanghai, China). All analytical-grade reagents, including anhydrous sodium acetate (NaOAc), anhydrous magnesium sulfate (MgSO_4_), sodium chloride (NaCl) and anhydrous sodium sulfate (Na_2_SO_4_), were supplied by Aladdin Biochemical Technology Co., Ltd. (Shanghai, China). C18, PSA, multi-walled carbon nanotubes (MWCNTs, purity > 98 wt%), -COOH functionalized MWCNTs (MWCNTs-COOH, 10–30 μm, purity > 98 wt%) and -OH functionalized MWCNTs (MWCNTs-OH, 10–30 μm, purity > 98 wt%) were provided by ANPEL Laboratory Technologies (Shanghai, China).

### 2.2. HPLC Conditions

An agilent 1290 HPLC system, which contains a photodiode-array detector (DAD) and a Zorbax XDB C18 column (4.6 mm × 250 mm, 5 μm), was used for separation. The solvent system containing A (acetonitrile) and B (10 mM ammonium acetate buffer solution, pH 4.5 adjusted with acetic acid) was pumped at 1.0 mL/min at 30 °C. A chromatographic gradient was applied and presented as follows: 0–5.0 min, 15% A; 5.0–5.1 min, with A increased from 15% to 20% gradually; 5.1–11 min, 20% A; 11.0–11.1 min, with A increased from 20% to 30% gradually; 11.1–23.0 min, 30% A. The injection volume was 20 µL. A wavelength of 258 nm was used. The typical chromatograms of the standard solution, spiked real sample and blank real sample are displayed in Appendix A. The retention time of each HAA was 5.85 min, 10.13 min, 14.42 min, 15.40 min and 19.50 min.

### 2.3. Preparation of Fe_3_O_4_-MWCNTs

Fe_3_O_4_-MWCNTs were prepared via a solvothermal method with slight modification [22]. Briefly, 100 mg of MWCNTs were dispersed in 50 mL of ethylene glycol, with bath ultrasonication for 120 min to form a homogeneous solution. Amounts of 0.675 g of FeCl_3_·6H_2_O and 0.5 g of polyethylene glycol were placed into the above black solution and sonicated for another 20 min. Subsequently, 1.8 g of NaOAc was added under magnetic stirring for 10 min. The obtained mixture was heated for 10 h at 200 °C in a 100 mL Teflon-lined stainless steel autoclave. After reaction, the black precipitate was collected by a permanent magnet, followed by washing several times with water and ethanol to remove all of the unreacted chemicals and impurities. Black solid materials were obtained after drying at 50 °C.

### 2.4. Sample Preparation

All braised sauce beef samples were collected from local deli counters, and thoroughly minced with a kitchen blender to obtain a homogeneous powder. An aliquot (2.5 g) of the braised sauce beef powder was added to a 50 mL centrifuge tube, which contained 7.5 mL of deionized water and 10 mL of extraction solvent (ACN containing 1% ammonium hydroxide), and vortexed violently for 1 min for extraction. Then, 2.0 g of MgSO_4_ and 0.5 g of NaOAc were added, followed by vortexing for 60 s. After centrifugation (1 min, 4000× *g*), the acetonitrile layer was obtained for further purification.

For the clean-up procedure, the acetonitrile supernatant (6 mL), together with Fe_3_O_4_-MWCNTs (35 mg), were placed into another 10 mL centrifuge tube. After vortexing for 60 s, the clean-up agent was magnetically separated from the sample solution. After separation, a 2 mL aliquot of the purified extract was collected and blown with nitrogen to remove the solvent. The residue was redissolved in methanol before HPLC analysis.

For comparison, the traditional QuEChERS process with C18, PSA or the mixture of C18 and PSA (C18 + PSA) as clean-up adsorbents was additionally provided.

### 2.5. Gravimetric Measurement of Co-Extracts

There was 20 mL of acetonitrile extracts, with or without purification through the procedure of Section 2.4, taken into 50 mL glass flasks that were previously weighed. Then, the acetonitrile solvent was removed by blowing with nitrogen, and the residue was accurately weighed. The weight difference was utilized to evaluate the clean-up efficiency of matrix co-extracts using different clean-up adsorbents. All of the experiments were carried out in triplicate, and the results were presented as means ± standard deviations.

### 2.6. Method Validation

Method validation was conducted in accordance with SANTE guidelines (SANTE/12682/2019) [23] through the following parameters: linearity, limits of detection (LODs), limits of quantitation (LOQs), precision and accuracy.

For the evaluation of the linearity, the blank braised sauce beef samples were spiked with seven different concentrations, and then extracted using the procedure of Section 2.4. The HPLC peak areas were plotted against the respective HAA concentrations to construct the working curves. The linearity of each target compound was assessed according to the correlation coefficient (*R*). For target analytes, LOD and LOQ were experimentally determined through the HPLC analysis of serial dilutions of the mixed standard solution to reach signal-to-noise ratios (S/N) ≥ 3 and (S/N) ≥ 10, respectively [24]. Meanwhile, the LOQ corresponded to the minimum concentration of HAAs in braised sauce beef that met the criterion, with recoveries between 70% and 120% and RSD ≤ 20% [23,25]. The accuracy was tested with recovery experiments that were carried out at high, middle, and low concentrations (500, 300 and 100 ng/g), with six replicates for each level. The intra- and inter-day precisions were tested by analyzing blank spiked samples at three different levels on one day and three consecutive days, respectively. The RSDs of the intra- and inter-day tests represented the precision.

## 3. Results and Discussion

### 3.1. Characterization of Fe_3_O_4_-MWCNTs

In our study, Fe_3_O_4_-MWCNTs were obtained using the solvothermal method with no need for nitrogen protection, which simplified the preparation process. Moreover, Fe_3_O_4_-MWCNTs obtained by this method possessed more stable properties. Transmission electron microscopy (TEM, JEM2100, Tokyo, Japan) was employed to investigate the morphology of the Fe_3_O_4_-MWCNTs, and the TEM image is demonstrated in Appendix A. It could be observed from Appendix A that Fe_3_O_4_ nanoparticles with nearly spherical shapes were scattered on the MWCNTs’ surface, which indicated the successful preparation of Fe_3_O_4_-MWCNTs. Moreover, only fewer bonding sites of the surface of MWCNTs were occupied by Fe_3_O_4_ nanoparticles. Therefore, they had little effect on the adsorption performance of MWCNTs.

### 3.2. Optimization of the Extraction Process

#### 3.2.1. Selection of Extraction Solvent

Complete extraction of HAAs from the sample matrix plays an essential role in achieving acceptable recoveries. Therefore, an appropriate optimized extraction solvent is necessary. For the QuEChERS application, ACN was the most commonly used extraction solvent, since it is able to extract many compounds with different polarities and lower amounts of proteins, lipids and other lipophilic co-extractives [26]. Previous studies have revealed that adding ammonium hydroxide or acetic acid to ACN may receive a higher recovery of the analyte [27]. Hsiao et al. (2017) found that the acidic acetonitrile could not extract IQ [4,5-b] from meat when DAD was used as the detector [12]. Therefore, we compared the recoveries of HAAs extracted using ACN, ACN containing 1% acetic acid and ACN containing 1% ammonium hydroxide. The rest of the conditions of the experiment were as follows: water volume, 5 mL; extraction salt, MgSO_4_/NaOAc; salt combination amount, 2.0 g/0.5 g; extraction time, 3 min; centrifugation time, 3 min; Fe_3_O_4_-MWCNTs amount, 35 mg/6 mL. As depicted in Figure 1A, adding ammonium hydroxide offered the best extraction efficiency for all of the analytes; most importantly, IQ [4,5-b] also achieved an acceptable recovery (81.6%) using DAD as the detector, which is an improvement over the research of Hsiao et al. (2017). Moreover, Yan et al. (2014) reported that HAAs with amphoteric character had high solubility in organic solvent under alkaline conditions, which may improve the recoveries of HAAs [28]. Given the overall results, ammonium hydroxide was selected as the solvent modifier of the extraction solvent for further extraction.

The content of ammonium hydroxide (NH_4_OH) in acetonitrile also has an important influence on the extraction efficiency during pretreatment, since HAAs have amphoteric properties [26]. The influence of ammonium hydroxide addition (0.5%, 1%, 3% and 5%, *v/v*) on the recoveries of HAAs was evaluated. The rest of the conditions of the experiment were as follows: water volume, 5 mL; extraction salt, MgSO_4_/NaOAc; salt combination amount, 2.0 g/0.5 g; extraction time, 3 min; centrifugation time, 3 min; Fe_3_O_4_-MWCNTs amount, 35 mg/6 mL. As depicted in Figure 1B, all of the recoveries extracted using ACN containing 1% ammonium hydroxide were within acceptable limits (70%–120%), and were a little higher than that of 0.5% ammonium hydroxide. It can be noted from Figure 1 that IQ had a higher overall recovery than those of the other four HAAs, which may be attributed to the fact that IQ had the weakest interaction with the matrix and is easier to be extracted. In addition, this phenomenon may also be related to matrix effect. Continuous increases in ammonium hydroxide percentages would bring out unacceptable recovery changes (>120%) for IQ.

Thus, the extraction solvent selected was acetonitrile with 1% ammonium hydroxide.

#### 3.2.2. Selection of the Volume of Water

Since the QuEChERS procedure was originally designed to analyze the fruit and vegetable samples with higher water content, it is commonly recommended that adding water to the samples, which contain a small amount of water, could improve the extraction efficiency [29]. This may be due to the fact that the added water would make the target analytes in the samples more accessible to ACN. The influence of the water volume was evaluated in our research by increasing the volume from 0 to 10 mL. The rest of the conditions of the experiment were as follows: extraction solvent, ACN containing 1% ammonium hydroxide; extraction salt, MgSO_4_/NaOAc; salt combination amount, 2.0 g/0.5 g; extraction time, 3 min; centrifugation time, 3 min; Fe_3_O_4_-MWCNTs amount, 35 mg/6 mL. As depicted in Figure 1C, 7.5 mL was sufficient to achieve the best extraction efficiency. Whereas, no significant difference in the recoveries of HAAs was observed over 7.5 mL. As a consequence, 7.5 mL of water was finally selected for the subsequent process.

#### 3.2.3. Selection of the Type of Extraction Salt

In a QuEChERS method, another essential factor focused on the selection of an appropriate combination of extraction salts, which could induce phase separation between the organic layer and water through a higher salting-out effect. Furthermore, extraction salt reduces the solubility of the analytes in water, and facilitates their partitioning into the organic phase, thus obtaining a higher extraction efficiency [30,31]. In this study, three groups of combined salts were investigated: (a) 2.0 g of MgSO_4_ and 0.5 g of NaOAc; (b) 2.0 g of Na_2_SO_4_ and 0.5 g of NaCl; (c) 2.0 g of MgSO_4_ and 0.5 g of NaCl. The rest of the conditions of the experiment were as follows: extraction solvent, ACN containing 1% ammonium hydroxide; water volume, 7.5 mL; extraction time, 3 min; centrifugation time, 3 min; Fe_3_O_4_-MWCNTs amount, 35 mg/6 mL. After adding the above salt combinations to the extract solution, the upper organic phase was collected and analyzed to compare the extraction recovery. As presented in Figure 1D, the salt combination of MgSO_4_/NaOAc provided slightly higher recoveries than the other salt combinations, while no significant differences were observed. According to Hsiao et al. (2017), the combination of MgSO_4_/NaOAc was frequently used in the QuEChERS procedure. As a result, we decided to choose the combination of MgSO_4_/NaOAc as the extraction salt.

Next, the influence of the amounts of the salt combination of MgSO_4_/NaOAc on recoveries was further assessed, and the amounts were designed as follows: (1) 1.0 g/0.25 g; (2) 2.0 g/0.5 g; (3) 3.0 g/0.75 g; (4) 4.0 g/1.0 g; (5) 5.0 g/1.25 g. The mass ratio of MgSO_4_ to NaOAc was fixed at 4:1. The rest of the conditions of the experiment were as follows: extraction solvent, ACN containing 1% ammonium hydroxide; water volume, 7.5 mL; extraction salt, MgSO_4_/NaOAc; extraction time, 3 min; centrifugation time, 3 min; Fe_3_O_4_-MWCNTs amount, 35 mg/6 mL. As presented in Figure 1E, the recoveries of all five analytes reached the maximum when the amounts of the combination of MgSO_4_/NaOAc were 2.0 g/0.5 g, while they rapidly decreased when the amounts were higher than 3.0 g/0.75 g. A portion of salts remained undissolved, and the volume of the acetonitrile layer was significantly reduced with a further increase in salt addition. We deduced that excessive addition of salts was not beneficial to phase separation, resulting in a marked decrease in the recoveries of target analytes. Thus, the selected amounts of MgSO_4_/NaOAc were 2.0 g/0.5 g.

#### 3.2.4. Selection of Extraction Time

The extraction time has a critical influence on the mass transfer process between the target analytes and extraction solvent, and thus affects the extraction recovery [32]. The vortex extraction time from 1 to 9 min was evaluated. The rest of the conditions of the experiment were as follows: extraction solvent, ACN containing 1% ammonium hydroxide; water volume, 7.5 mL; extraction salt, MgSO_4_/NaOAc; salt combination amount, 2.0 g/0.5 g; centrifugation time, 3 min; Fe_3_O_4_-MWCNTs amount, 35 mg/6 mL. The results displayed in Figure 1F indicated that satisfactory recoveries could be achieved when the extraction time was 1 min. It was concluded that 1 min was sufficient to achieve thorough interaction between extraction solvent and the target analytes, while longer extraction times displayed no obvious change in the extraction yield of HAAs. Therefore, 1 min was employed for subsequent studies. Lai et al. (2023) reported that the shaking time after adding extraction solvent in the traditional QuEChERS method was 10 min [14]. Compared with this reference, the extraction time of this method was significantly shortened, which was conducive to shortening the whole sample pretreatment process. It can be attributed to the fact that large dosages of adsorbents (300 mg PSA, 900 mg MgSO_4_ and 300 mg C18EC in 6 mL supernatant) in the traditional QuEChERS approach would take a long time to achieve complete extraction.

#### 3.2.5. Selection of Centrifugation Time

The step of centrifugation is essential for completing the solid–liquid separation. Furthermore, centrifugation time also influences the volume of the upper organic layer [30]. Thus, an appropriate centrifugation time is vital for improving the extraction recovery. The influence of varying the centrifugation time from 1 to 9 min was evaluated. The rest of the conditions of the experiment were as follows: extraction solvent, ACN containing 1% ammonium hydroxide; water volume, 7.5 mL; extraction salt, MgSO_4_/NaOAc; salt combination amount, 2.0 g/0.5 g; extraction time, 1 min; Fe_3_O_4_-MWCNTs amount, 35 mg/6 mL. As demonstrated in Figure 1G, satisfactory recoveries were obtained at 1 min, and the recoveries changed slightly when the centrifugation time was further prolonged. Therefore, 1 min was used in this experiment. The observed centrifugation time was significantly shorter than that reported in the literature of Lai et al. [14] (10 min), which was conducive to shortening the whole sample pretreatment process.

### 3.3. Optimization of the Clean-Up Process

#### 3.3.1. Selection of the Adsorbent

Compared with traditional adsorbents PSA and C18, MWCNT materials have become a research hotspot in the field of clean-up adsorbents in recent years [17,31]. In this study, the clean-up performance of magnetic MWCNTs (Fe_3_O_4_-MWCNTs, Fe_3_O_4_-MWCNTs-COOH and Fe_3_O_4_-MWCNTs-OH) was compared with traditional adsorbents, including C18 and PSA, as well as a mixture of PSA and C18 (PSA + C18). In the purification process, 7.5 mg of each type of magnetic MWCNT materials was added to 1 mL acetonitrile extracting solution of braised sauce beef, while 50 mg/mL were added for PSA or C18, and 100 mg/mL for the mixture of PSA and C18 (50 mg PSA and 50 mg C18). From the recovery results of Figure 2A, compared with the other clean-up adsorbents, the extracting solution purified using Fe_3_O_4_-MWCNTs displayed better or equivalent recoveries.

The clean-up efficiency in removing matrix co-extracts of the above-mentioned clean-up materials was further assessed using gravimetric analysis. As displayed in Figure 3, 7.3 mg of the remaining co-extracts was obtained with purification by Fe_3_O_4_-MWCNTs, while 16.5 mg was obtained without purification, suggesting that Fe_3_O_4_-MWCNTs was efficient in removing matrix co-extracts. Fe_3_O_4_-MWCNTs-COOH and Fe_3_O_4_-MWCNTs-OH provided superior removal abilities of matrix co-extracts than Fe_3_O_4_-MWCNTs, but the recovery of IQ [4,5-b] was less than 60% when they were used as the adsorbents. The reason for this phenomenon may be that the hydroxyl or carboxyl groups on the MWCNT surfaces strengthened the interaction of the matrix with the target compounds. In comparison, C18 possessed similar matrix removal efficiency to Fe_3_O_4_-MWCNTs, while its recoveries were lower than those of Fe_3_O_4_-MWCNTs. The matrix removal ability of PSA was the lowest. The combination of PSA and C18 displayed the largest clean-up efficiency, but their recoveries were lower than those of Fe_3_O_4_-MWCNTs. Besides, the combination of two clean-up materials made the pretreatment process troublesome, and the centrifugation process (3–10 min), which is needed to separate the adsorbents from the sample solution, was time-consuming.

Last but not least, when Fe_3_O_4_-MWCNTs were used as adsorbents, the solid–liquid separation could be completed within a few seconds by an external magnet without any additional centrifugation process. Moreover, compared with the combined effect of multiple adsorbents, only one adsorbent (Fe_3_O_4_-MWCNTs) used in the QuEChERS method made sample pretreatment easier and more time-saving, and facilitated high throughput analysis. Furthermore, the preparation of Fe_3_O_4_-MWCNTs was accomplished using a simple method and inexpensive MWCNTs. Considering their advantages of efficiency, speed, and simplicity as clean-up adsorbents, Fe_3_O_4_-MWCNTs could be used to analyze multiple targets in complex samples.

#### 3.3.2. Selection of the Amount of Adsorbent

In order to obtain satisfactory recovery and good clean-up performance, the amount of Fe_3_O_4_-MWCNTs (5, 15, 25, 35 and 45 mg for 6 mL of the extract) was investigated. The rest of the conditions of the experiment were as follows: extraction solvent, ACN containing 1% ammonium hydroxide; water volume, 7.5 mL; extraction salt, MgSO_4_/NaOAc; salt combination amount, 2.0 g/0.5 g; extraction time, 1 min; centrifugation time, 1 min. As described in Figure 2B, the recovery of Phe-P-1 was highest when the amount increased from 5 mg to 15 mg, but remained almost unchanged with a further amount increase. For the other four target compounds, all the recoveries were in satisfactory ranges (70–120%) with the increase of the amount of Fe_3_O_4_-MWCNTs, while, when it was over 25 mg, the recoveries decreased to some extent. Additionally, both the gravimetric test and the comparison of color of the purified solution were also performed to study the effect of Fe_3_O_4_-MWCNT amounts on the clean-up performance. The results demonstrated that the reduction of the co-extract weight became slow when the Fe_3_O_4_-MWCNT amounts exceeded 35 mg (Appendix A) and the final extract after purification with 35 mg of Fe_3_O_4_-MWCNTs was colorless and transparent (Appendix A). Given the overall results, the amount of Fe_3_O_4_-MWCNTs was selected as 35 mg/6 mL. Compared with the traditional QuEChERS method in Lai et al. (2023) in which the amounts of clean-up adsorbents were 300 mg PSA, 900 mg MgSO4 and 300 mg C18EC in 6 mL supernatant [14], the modified QuEChERS approach had less usage of adsorbent. Moreover, the cost of Fe_3_O_4_-MWCNTs is low (approximate $ 0.13 per 35 mg of adsorbent). Thus, the method possesses the advantage of having a low cost per sample.

### 3.4. Method Validation

The results of parameter validation are displayed in Table 1. All five target analytes had satisfactory linearity, with *R* values varying from 0.9983 to 0.9994. According to signal/noise ratios of 3 and 10, the LOQs and LODs for all five HAAs were 4.2–12.0 ng/g and 3.0–4.2 ng/g, respectively. The obtained LOQ value was lower than the prescribed maximum residue limit (MRL) recommended by European Union legislation, which sets the value at 50 μg/kg [33].

The method accuracy and precision were evaluated in terms of recoveries and the RSDs of the intra- and inter-day tests. According to the experimental results, the recoveries of the three concentration levels ranged from 78.5% to 103.2%. The intra-day and inter-day RSDs ranged between 0.6–4.6% and 3.5–4.6%, respectively. The analytical results of accuracy and precision were within the validation criteria (recovery between 70 and 120% with RSD ≤ 20%) [23]. Compared with previous literature, Hsiao et al. used LC–MS-MS analysis and a QuEChERS method with a mixture of PSA and C18 as adsorbents to extract HAAs from meat, and achieved a recovery of 58.9–117.4% with RSD less than 25.68% [12]. Jinap et al. conducted HPLC-DAD analysis on beef and chicken satay, employing Oasis MCX cartridge purification; they obtained recoveries of 43–92% and 49–98%, respectively [34]. The obtained recovery and RSD values were higher than, or comparable to, those achieved by the reported methods in the aforementioned literature, implying that the modified QuEChERS technique possessed excellent accuracy and precision.

### 3.5. Comparisons with Other Studies

The offered technique in this study was further compared with two reported techniques, namely, a frequently used technique (SPE) and a popular technique in recent years (QuEChERS). From the reported reference data in Table 2, the lowest recovery of the present method was obtained as 78.5%, compared to 28.9%, 43.0%, 49.0%, 56.2%, 52.3%, 58.9% and 42.5% as the corresponding values from other reported references. The precision RSD (<4.6%) obtained in this study was lower than those reported in all references in Table 2. This shows that this method displayed better or equivalent accuracy and precision. The LOD for the present method conforming to the criteria was slightly lower than those obtained with MS detector, but was acceptable. Meanwhile, the total sample pretreatment time for one sample required approximately 4 min, much shorter than that of the traditional QuEChERS technique using a mixture of C18 and PSA as adsorbents, and SPE methods. This meant that when Fe_3_O_4_-MWCNTs were used as the clean-up adsorbent, shorter extraction and centrifugation times in the extraction process coupled with magnetic separation in the clean-up process simplified the sample pretreatment process and made the separation more time-saving. In addition, the current method required the minimum usage of clean-up adsorbent compared with the traditional QuEChERS approach in [12]. The cost of Fe_3_O_4_-MWCNTs is low (approximate $ 0.13 per 35 mg of adsorbent). The comparison further proves that the modified QuEChERS method of applying Fe_3_O_4_-MWCNTs to extract HAAs possessed obvious superiorities in high precision and accuracy, speed, time saving and low cost per sample. 

### 3.6. Analysis of Real Samples

Seventeen real samples of braised sauce beef (numbered Beef 1 to Beef 17) were obtained from different deli counters and analyzed using the above proposed method. As can be seen from Appendix A, three analyzed HAAs (IQ, Harman and Norharman) were detected in Beef 1, Beef 8, Beef 13 and Beef 14. This was in accordance with the study of Yao et al. [39], who reported that IQ, Harman and Norharman were also detected from braised sauce beef; however, all of their concentrations were below the LOQ in our study. Only IQ was detected in Beef 5 at a concentration below the LOQ. Both Phe-P-1 and IQ [4,5-b] were not detected during our research, and none of the analyzed HAAs were detected in the other 12 samples.

The differences in HAAs in both amount and type among the different samples may be related to multiple factors. Lan et al. [40] reported that the addition of soy sauce and sugar in the marinating juice could facilitate the formation of HAAs in marinated pork, since sugar and soy sauce rich in amino acids were important precursors. Furthermore, according to Yao et al. [39], the HAA amounts in the chicken braised in soup cycled 20 times was approximately 7 times higher than that braised in soup cycled 5 times; moreover, the HAA content increased linearly with the cooking cycles of the marinating juice. Therefore, to clarify these differences, further research needs to be carried out in two aspects: (a) the analysis of HAAs in the marinating juice from different deli counters; (b) the factors affecting the formation of HAAs, including cooking method, cooking cycles of the marinating juice, and the type and quantity of seasonings and spices added to the marinating juice. Relevant studies are ongoing.

## 4. Conclusions

This report developed a modified QuEChERS technique based on Fe_3_O_4_-MWCNTs to determine HAAs in braised sauce beef samples combined with HPLC-DAD detection. Firstly, the Fe_3_O_4_-MWCNT materials were successfully prepared with a simple solvothermal method. Then, the conditions of the extraction process were optimized to comprise 10 mL of ACN containing 1% ammonium hydroxide, 7.5 mL of water, 2.0 g of MgSO_4_ and 0.5 g of NaOAc, 1 min of vortex extraction time, and 1 min of centrifugation time. Then, a comparison with traditional adsorbents was carried out, and Fe_3_O_4_-MWCNTs displayed better or equivalent clean-up efficiency and experimental recoveries than PSA and C18. Considering the effect of the amount of Fe_3_O_4_-MWCNTs on clean-up performance and recoveries, 35 mg/6 mL was employed for optimal usage. The sample pretreatment for one sample was completed in about 4 min. Magnetic separation simplified the pre-processing operation, and made the separation more time-saving. Reduced usage of Fe_3_O_4_-MWCNTs provided the method another advantage of having a low cost per sample. Under optimized conditions, this method had good accuracy and precision, and acceptable LOQs and LODs. The recoveries were between 78.5% and 103.2%, and the precision was lower than 4.6%. The LODs of the method were 3.0–4.2 ng/g. Finally, the method was also applied to analyze 17 real braised sauce beef samples, and three targets (IQ, Harman and Norharman) were detected under the LOQ in 5 samples. In general, the proposed method, with its simplicity, time savings and low cost per sample, could be suitable for the routine monitoring of HAAs in braised sauce beef samples; moreover, Fe_3_O_4_-MWCNTs could be used as an alternative adsorbent in the QuEChERS method.

## Figures and Tables

**Figure 1 foods-12-00138-f001:**
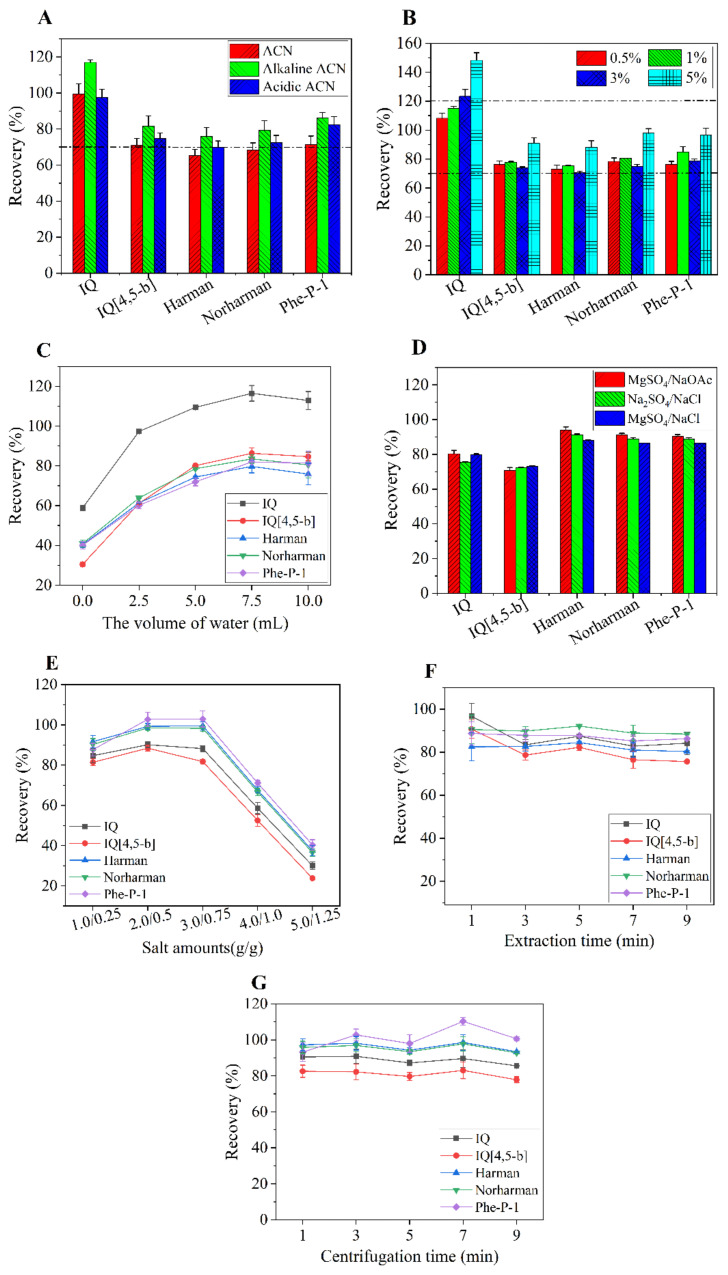
Effects of experimental conditions of the extraction process on the recovery. (**A**): acetonitrile, alkaline acetonitrile and acidic acetonitrile; (**B**): different percentages of ammonium hydroxide in acetonitrile; (**C**): volume of water; (**D**): type of extraction salt; (**E**): amount of MgSO_4_/NaOAc; (**F**): extraction time; (**G**): centrifugation time. Spiked concentration, 300 ng/g.

**Figure 2 foods-12-00138-f002:**
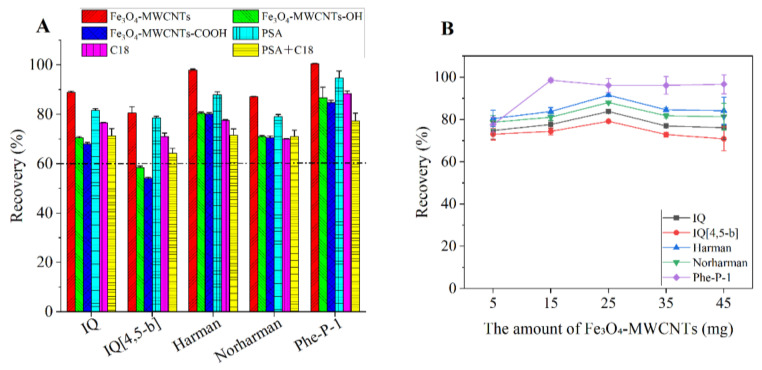
Effects of important factors of the clean-up process on the recovery. (**A**): type of adsorbent; (**B**): amount of Fe_3_O_4_-MWCNTs. Spiked concentration, 300 ng/g.

**Figure 3 foods-12-00138-f003:**
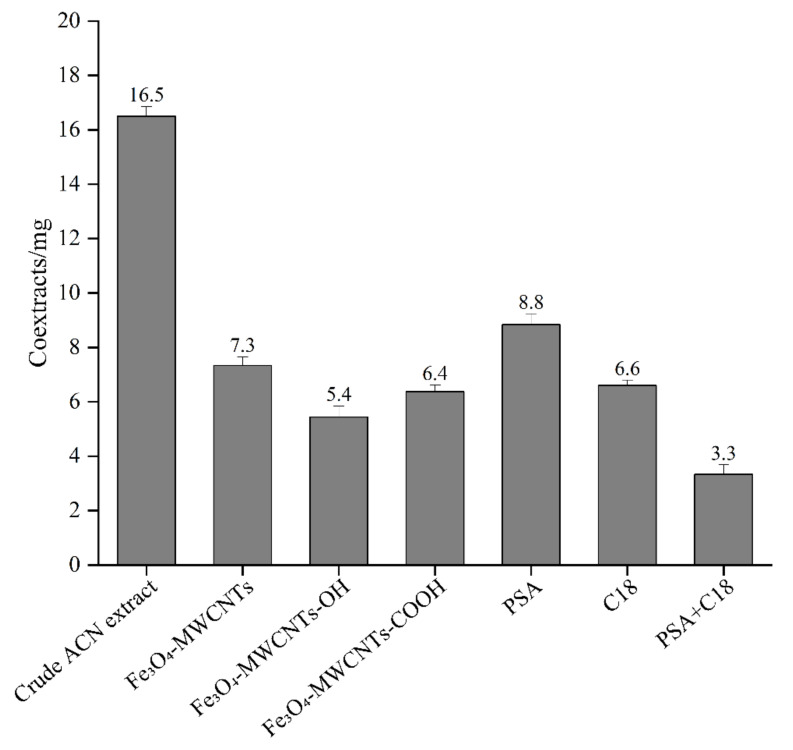
Amount of matrix co-extracts obtained through gravimetric measurement after purification with different clean-up materials (*n* = 3).

**Table 1 foods-12-00138-t001:** Analytical performances of the method.

Analytes	Linear Range (μg/g)	*R*	Recovery ± RSD (%, *n* = 6)			Inter-Day RSD (%)	LOQs (ng/g)	LODs (ng/g)
			Low (100 ng/g)	Middle (300 ng/g)	High (500 ng/g)			
IQ	0.012–12.0	0.9989	82.8 ± 4.6	82.1 ± 3.2	82.3 ± 3.1	3.9	4.2	3.0
IQ [4,5-b]	0.012–12.0	0.9994	78.6 ± 2.9	78.8 ± 4.4	78.5 ± 2.5	4.6	12.0	4.2
Harman	0.012–12.0	0.9986	90.9 ± 3.8	89.6 ± 3.5	89.6 ± 3.3	4.1	4.2	3.0
Norharman	0.012–12.0	0.9991	89.1 ± 3.5	88.1 ± 3.4	88.2 ± 3.3	4.4	4.2	3.0
Phe-P-1	0.012–12.0	0.9983	103.2 ± 2.8	94.9 ± 0.6	101.6 ± 4.4	3.5	12.0	4.2

**Table 2 foods-12-00138-t002:** Comparison of the present method with other methods.

Instrument	Extraction Method	Food Samples	Recovery (%)	RSD (%)	LOD	Sample Pretreatment Time (min)	Amounts of Adsorbent (mg/6 mL)	Ref.
UHPLC–APCI–MS/MS	SALLME ^1^ extraction/SPE purification	Cooked beef	60.7–108.5	<25.1	1–5 pg/µL	>100	-	[11]
LC–MS-MS	QuEChERS	Meat	58.9–117.4	<25.6	0.003–0.05 ng/mL	32	300 mg PSA + 900 mg MgSO_4_ + 300 mg C18EC	[12]
HPLC-DAD	SPE ^2^	Chicken/beef	43.0–92.049.0–98.0	-	0.3529–1.549 ng/g	>180	-	[34]
UPLC–MS/MS	Extrelut-SPE	Roasted beef patties	56.2–106.1	<11.2	0.008–0.053 ng/g	68.6	-	[35]
UHPSFC–MS/MS	SPE	Grilled meals/fish	52.3–97.5	<6.0	0.01–0.05 µg/kg	-	-	[36]
UHPLC–MS/MS	Acetonitrile/SPE	Meat products	42.5–99	<12.2	0.007–0.202 ng/g	>50	-	[37]
HPLC-DAD	SPE	Cooked meatballs	68.9–87.8	-	0.04–1.40 ng/g	>60	-	[38]
HPLC-DAD	Modified QuEChERS	Braised sauce beef	78.5–103.2	<4.6	3.0–4.2 ng/g	4	35	This study

^1^ Salting-out liquid–liquid microextraction; ^2^ solid-phase extraction.

## Data Availability

Data are contained within the article and Appendix A.

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
