# Peer review of "Development of a Modified QuEChERS Method Based on Magnetic Multi-Walled Carbon Nanotubes as a Clean-Up Adsorbent for the Analysis of Heterocyclic Aromatic Amines in Braised Sauce Beef"

_foods, 2022, doi:10.3390/foods12010138_

Round 1

Reviewer 1 Report

Comments to authors

The study is interesting. However the manuscript must be improved.

Specific comments

-          In my opinion the abstract shall be revised highlight the main findings of the stud, especially L.14-20 that are very generic.

-          L.47-57: Are presenting the history of QuEChERS. This paragraph shall be reduced only to the necessary part and connected to scope of this research, especially with the next paragraph (L.58-68). In relation to paragraph 2.4, the normal sorbents used in QuEChERS shall be presented and explained (functionality, expected effect).

-          Introduction shall contain existing analytical methodologies and sample preparation applied for  HAAs, and especially to braised beef meat or meat more general. The state-of-the-art related to this research must clearly presented. Please some indicative references (and not exhaustive) below:

o   Gizem et al. 2023 Food Chemistry 133919: A review of the currently developed analytical methods for the determination of biogenic amines in food products.

o   Giangming et al. 2023, Food Chemistry, 134822: Heterocyclic aromatic amines in roasted chicken: Formation and prediction based on heating temperature and time.

o   Lai et al.,2023, Food Chemistry, 134291: Extraction of heterocyclic amines and polycyclic aromatic hydrocarbons from pork jerky and the effect of flavoring on formation and inhibition.

o   Feng et al. 2022, Food Control: Improved enrichment and analysis of heterocyclic aromatic amines in thermally processed foods by magnetic solid phase extraction combined with HPLC-MS/MS.

-          L. 82: The abbreviation for IQ and IQ[4,5] is not presented beforehand. Please provide.

-          MATERIALS AND METHODS: There is no description nor presentation of the method  validation (reference, criteria, methodology etc.).

-          L.163-165: The authors shall provide an explanation (even tentative) why this effect is occurring. Isn’t due to the amphoteric character of amines? Please amend discussion.

-          In order to assess the recoveries of this analytical method, performance criteria, together with a reference document, shall be provided.

-          In general IQ (Figure 1) presents the highest recoveries while the other 4 HAAs. Please amend discussion and provide an explanation for this phenomenon.

-          The authors are presenting their results, which are interesting, but there is no discussion compared to other existing methodologies, either analytical or sample preparation. This discussion will certainly enhance the document.

-          My major comment is for Paragraph 3.4. The authors are presenting very fast, and with no discussion their results. This part requires significant revision.

o   The provide LOD and LOQ. How were they calculated? Based on which methodology? Comparison with other methods? What is the added value compared to other analytical methods?

o   Table 1: Provides a low, middle and high concentration. To What concentrations these levels are referring to? Please provide numerical values.

o   Table 1: The authors provide a LOD from 4.2-12.0 ng/g, while the linear range is from 0.012-12 ng/g.

o   Table 1: The authors provide LODIQ of 3.0 and LOQ of 4.2 ng/g. How LOD and LOQs were calculated? The same is valid for Harman, Noharman. Please provide an explanation and provide the reference for calculating LODs and LOQs.

-          Table 2 highlights that UPLC-MS/MS is significantly more sensitive with the presented method. Therefore what is the added value of this method?

-          Furthermore, SPE with HPLC-DAD (see Ref. 32 and 37; Table 2) seems to have lower LOD compared to the current method. Please discuss and explain the added value of the current method.

-          Based on the provided LODs/LOQs, what is the regulatory framework in order this method to be considered as sufficient? Please amend the legislative framework (if exists)

-          Based on my previous comment, the main overview coming from Table 2 shall be introduced to the “Introduction” part.

-          L.350: the claim “not detected” or “not found” is misleading. It shall be reviewed to “below LOD”! HAAs might be present but to lower concentrations.

-          Conclusions is very generic. The main conclusion from this study shall be provided, including the preparation of the MWCNTs and the parameters assessed and evaluated during the method development/validations.

General comments

It would be nice to provide the chemical 

Reviewer 2 Report

This manuscript is original and has relevance to the outreach of Foods. The manuscript has an adequate reference list, which is well processed throughout the study. The definition, justification and fulfillment of the objective is worked on.

The modified QuEChERS method is optimized, validated and applied to the analysis of real braised beef samples. However, I have found some points in the manuscript, which need to be corrected.

- I think it is necessary to modify the title, clarifying that the use of MWCNTs in the cleanup stage is proposed. As an example: "Development of a QuEChERS method based on multi-walled magnetic carbon nanotubes as a clean-up adsorbent for the analysis of heterocyclic aromatic amines in braised beef"

- There are other important HAAs and these have been determined in the samples analyzed. Why did the authors select only these five HAAs? It is necessary to clarify this at work.

- Reagents and Materials: The acronym NaOAc is not indicated in this section.

- HPLC conditions: Specify the retention time of each HAA, show a comparative chromatogram of a real sample with and without the application of the proposed extraction and clean-up method.

- Line 148-149: The authors state "In addition, the Fe3O4-MWCNTs obtained by this method had more stable properties." What studies were done Expand this information.

- Optimization of the Extraction Process:

Several parameters were studied. The optimal values of the different parameters are selected one by one. Why was the experimental design not applied?

Specify the rest of the conditions of the experiment in the study of each variable of the proposed methodology. Pay special attention to the type and quantity of salts used in the QUECHERs stage, why are the recoveries (%R) shown in Figure 1 B generally higher than those shown in Figure 1 D?

The authors present the methodology with advantages of low cost and time saving. In this sense, is the cost of Fe3O4-MWCNTs lower than C18 and PSA?

On the other hand, it would be interesting to study the R(%) at times less than 1 min in the vortex and centrifugation stages. This can be an additional advantage in saving time of the method proposed in this work.

The total sample treatment time for each cited method can also be added to Table 2, thus highlighting the methodology proposed by the authors in this paper.

- Validation of the method: What concentrations were added to the samples? Specify the values of the Low, Medium and High concentrations in table 1

Round 2

Reviewer 1 Report

The authors have addressed my comments and therefore I recommend the publication of this scientific work.

Reviewer 2 Report

The authors have satisfactorily responded to my questions and made the necessary changes to the manuscript,  therefore I recommend its publication.